# The Effect of the Iridium Alloying and Hydrogen Sorption on the Physicochemical and Electrochemical Properties of Palladium

**DOI:** 10.3390/ma16134556

**Published:** 2023-06-24

**Authors:** Katarzyna Hubkowska, Małgorzata Pająk, Andrzej Czerwiński

**Affiliations:** Faculty of Chemistry, University of Warsaw, Pasteura 1, 02-093 Warsaw, Poland; mpajak@chem.uw.edu.pl (M.P.); aczerw@chem.uw.edu.pl (A.C.)

**Keywords:** hydrogen storage, palladium-iridium alloy, electrodeposition, hydrogen absorption, electrochemical dissolution, α → β

## Abstract

Thin layers (up to 1 µm) of Pd-Ir alloys were electrodeposited from aqueous, galvanic baths of PdCl_2_ and IrCl_3_ mixtures. The morphology of the electrodeposits was examined by means of scanning electron microscopy. The composition of alloys was determined with the use of energy-dispersive spectroscopy, atomic absorption spectrometry, X-ray photoelectron spectroscopy, and Auger electron spectroscopy. For the studies of the electrochemical properties of alloys, cyclic voltammetry, chronoamperometry, and chronopotentiometry were used. It was found that Pd-Ir alloy electrodes were surface-enriched with Pd. Pd-Ir alloys subjected to different electrochemical treatment involving hydrogen sorption changed their surface state. The continuous hydrogen sorption enhanced the Ir ions’ dissolution. The values of thermodynamic functions of hydrogen sorption in strong alkaline electrolytes were comparable with those in acidic electrolytes, whereas the kinetics of the process in alkaline medium was hindered. The miscibility gap in the Pd-Ir-H system vanished for the electrode containing ca. 93.7 at.% Pd.

## 1. Introduction

The biggest challenge of the modern world is the production of “green” energy and finding a method for its safe storage. One of the most important, non-toxic, clean energy carriers is hydrogen. It is of interest in transportation, especially when used in hydrogen fuel cells or internal combustion engines [1,2,3,4]. However, the problem of efficient and safe storage of hydrogen is an essential obstacle to overcome. The most technologically relevant class of hydrogen storage materials are metal hydrides (MH_x_) [5,6]. Metal hydrides have a higher hydrogen storage density than gaseous or liquid hydrogen. Consequently, metal hydride storage is volume-efficient and safe. The research on metal hydride storage materials focuses on improving the volumetric capacities, reaction thermodynamics, and hydrogen adsorption/desorption kinetics of potential material candidates. Storage by absorption in the form of chemical compounds has definite advantages from the safety perspective. Bulk Pd is the only metal which can form a hydride at ambient temperature and hydrogen pressure, leading to solid-state storage under moderate conditions. Therefore, Pd and its alloys, due to the ability of hydrogen absorption and good electrochemical properties, are important in view of energy storage and conversion, especially as a model system to be utilized for the interpretation of the hydrogen sorption process in other hydrogen-absorbing materials. Whilst Pd and its binary or ternary alloys with other noble metals such as Pt, Rh, Ru, Ag, or Au were extensively studied and electrochemically characterized [7,8,9,10,11,12,13,14,15], in the literature there is a lack of information regarding electrodeposited Pd-Ir alloys, and their properties in terms of hydrogen electrosorption are still not thoroughly examined [16,17]. Alloying with transition metals is an efficient way to improve the utility of the Pd constituent [18]. Depending on the metals’ proportions, increased activity, selectivity, or electrocatalytic properties can be obtained. The Pd-Ir system has a large difference in reduction potential; in the case of Ir^3+^/Ir, it is 1.19 V, while for Pd^2+^/Pd the reduction potential equals 0.48 V [19]. Ir shows excellent stability in acidic medium due to its high reduction potential. Therefore, Pd-doping with Ir improves the durability of Pd alloys. Moreover, Ir modifies the electronic structure of Pd alloys. For this reason, Pd-Ir bimetallic alloys have found many chemical and physical applications, i.e., as electrocatalysts for oxygen reduction or hydrogen evolution reaction [18,19,20,21].

Pd and Ir are palladium group metals which form a strong de-mixing alloy system in the bulk [22,23]. Phase diagrams of bulk alloys are governed by their crystal and electronic structure and the size of the constituents. In the work of Raub [24], a wide miscibility gap in the Pd-Ir system at high temperatures is shown with the critical temperature of 1476 °C. In the case of this alloy, the thermodynamical equilibrium state is reached very slowly. It is explained by differences in the melting points of pure metals as their atomic radii and lattice constants show only small differences (Table 1).

The thermodynamic equilibrium of thin-film alloys may be strongly different compared to that of the bulk. New ordered phases may appear as a result of minimization of surface and interface strains and energy. Therefore, alloying between immiscible elements is allowed [25]. Pd-Ir constitutes face-centered cubic alloys with a reduced lattice constant compared to pure metals. The alloy lattice constant decreases with the increasing content of Ir, with a small negative deviation from Vegard’s law [16,17].

Studies of the hydrogen interaction with Pd and Ir are rather limited and revealed that Ir is able to adsorb hydrogen, while in the case of Pd adsorption, an absorption process occurs [22,26]. Therefore, the aim of this study was the electrochemical investigation of the Pd-Ir binary system over a wide composition range, obtained as thin electrodeposits. A detailed understanding of these systems is of interest due to their catalytic properties and because they form a model system for other hydrogen-absorbing materials. In our previous study [16], we showed the basic hydrogen sorption properties of Pd-Ir alloys, such as: the influence of the Pd bulk content on the α → β phase transition potential and the maximum absorption capacity in acidic medium. Moreover, X-ray diffraction studies were performed to confirm the creation of the contracted alloy. In the following sections, the detailed physicochemical and electrochemical studies on the Pd-Ir alloys are presented, with an emphasis on the influence of the hydrogen sorption on the surface/bulk properties of the electrodes. Furthermore, the process of hydrogen sorption in strong alkaline and acidic media was compared in view of its thermodynamics and kinetics.

## 2. Materials and Methods

### 2.1. Materials

Iridium(III) chloride hydrate (99.8%) and palladium(II) chloride (Premion®, 99.999%) were purchased from Alfa Aesar (London, UK). Gold wire (0.5 mm diameter, 99.99%) was obtained from Mint-Metals (Radzymin, Poland). All solutions were prepared in ultrapure water (conductivity 0.055 µScm^−1^) with the use of analytical-grade reagents.

### 2.2. Methods

#### 2.2.1. Sample Preparation

Pd-Ir alloys were electrodeposited at constant potential on Au wire from the mixtures of 0.11 M PdCl_2_ in 1 M HCl and 0.5 M IrCl_3_ by means of the CHI 760d electrochemical workstation (CH Instruments Inc., Austin, TX, USA). Alloys of various compositions were obtained by changing the galvanic baths’ compositions and the potential of deposition (details in point 3.1). The alloy composition is expressed as the atomic percentage.

#### 2.2.2. Sample Analysis

The bulk composition of the prepared Pd-Ir alloys was determined using atomic absorption spectrometry (AAS, Thermo Scientific 3300, Waltham, MA, USA) and energy-dispersive X-ray spectroscopy (EDS, Quantax 400, Bruker, Billerica, MA, USA; electron beam energy: 10–15 keV, spectrum acquisition time: 100–120 s, EHT: 15 kV, probe current 15 pA). The surface morphology of alloys was investigated with the use of scanning electron microscopy (SEM, Zeiss, Oberkochen, Germany; EHT: 3 kV; probe current: 30 pA; system vacuum: 1.8·10^−9^ bar, gun vacuum: 9.5·10^−13^ bar). For X-ray photoelectron spectroscopy (XPS) and Auger electron spectroscopy (AES), a Microlab 350 (Thermo Electron, Waltham, MA, USA) equipped with a FEG-tip (Field-Emission Electron Gun) and a twin anode source (AlK_α_ and MgK_α_) was used. XPS experiments were executed by an AlKα (hv = 1486.6 eV) anode X-ray source operated at 15 kV and an emission current intensity of 20 mA. The survey spectra and high-resolution spectra from the total surface area of 0.2 cm^2^ were recorded using 100 eV and 40 eV pass energy. Wagner sensitivity factors were used to estimate the quantitative chemical composition. The excitation energy used in the AES measurement was 10 kV. The sensitivity coefficients determined on the basis of the database implemented in the Thermo Avantage v.5.9911 software were used to estimate the quantitative chemical composition: PdMN1—0.612, OKL1—0.424, and IrMN3—0.843. Shirley’s background cutoff line was used to determine the peak intensities.

#### 2.2.3. Electrochemical Measurements

The electrochemical measurements were conducted in 0.5 M H_2_SO_4_ or 6 M KOH de-aerated with Ar (99.999%. Air Products, Warsaw, Poland). During the measurements, the stream of argon was directed above the solution. In the studies of hydrogen electrosorption, electrodes were cycled in the potential range of −0.15 to 0.5 V. These experiments were performed at the temperature of 298 K (and at temperatures of 283 K, 298 K, 313 K, and 328 K to determine the values of the thermodynamic functions) controlled by a Lauda RE 630 thermostat (Lauda, Lauda-Königshofen, Germany) in a three-electrode system. A Hg|Hg_2_SO_4_|0.5 M H_2_SO_4_ or Hg|HgO|6 M KOH electrode was used as the reference electrode, Pt gauze was the auxiliary electrode, and the working electrode was gold wire with alloy, electrodeposited as described in Section 2.2.1 (alloy thickness ca. 0.4–1 µm, height ca. 0.5 mm). The electrochemical tests were conducted with the use of the CHI 760d electrochemical workstation (CH Instruments Inc., Austin, TX, USA). All potentials in the presented work are referred to as RHE (or SHE for deposition potentials). Before collecting electrochemical data concerning the analysis of thermodynamics and kinetics, all prepared electrodes were subjected to a hydrogen pretreatment procedure (HPP) [27], which is a series of voltammetric and chronoamperometric runs in a hydrogen absorption/desorption potential range conducted until repeatable voltammograms were obtained.

## 3. Results

### 3.1. Electrodeposition of the Pd-Ir Alloys

Pd-Ir alloys of different compositions were obtained using the influence of the ion concentration and electrodeposition potential on the composition of the alloys. It was found that generally, a lower deposition potential and higher concentration of Ir ions in the deposition bath resulted in a higher Ir content in the alloy. However, the analysis of the results presented in Figure 1a,b reveals that the deposition potential rather than the concentration had a stronger impact on the increase of the Ir content in the alloys. Figure 1a presents the dependence of the Pd bulk content on the Ir to Pd ions’ concentration ratio. It can be concluded that even a significant change in the concentration of Ir ions in the deposition bath had a slight impact on the change of the composition of the alloys. In Figure 1b, it can be seen that the decrease of the deposition potential caused a linear increase of the Ir content. It is well-visible that for different deposition baths, the variation of the alloys’ composition was insignificant, especially for relatively high deposition potentials such as 0.35 V and 0.4 V. In the literature, there are some reports describing the strong influence of the alloying metal concentration on the deposition efficiency [13]. In the case of the Pd-Ir alloys, it can be concluded from Figure 1c that the increase of the Ir ions’ concentration in the deposition baths had a not unfavorable impact on the deposition current efficiency. For the Ir concentration in the range of 0.3 to 0.5 M, the deposition efficiency was equal to, on average, ca. 90 at.%. These results are similar to those obtained for the electrodeposited Pd-Rh alloys, where the deposition efficiency was in the range of 93% to 100 at.% [28].

### 3.2. The Study of the Morphology, Composition, and the Homogeneity of Pd-Ir Alloys

The quality of the electrodeposited films was studied with the use of SEM. Figure 2 presents the SEM images of three Pd-Ir alloys with various compositions, and pure Pd as well as pure Ir for comparison. The SEM results revealed that the electrodeposition of alloys from chloride baths containing different concentrations of Pd and Ir ions enabled to obtain free of cracks and compact deposits. After comparing the morphologies of the electrodeposits, it was noted that just after the addition of 0.5 at.% of Ir to Pd, the morphology of the samples significantly changed. However, the morphology of this sample more closely resembled the morphology of pure Pd than pure Ir. Surprisingly, after another addition of 0.5 at.% Ir, the morphology of the sample was more similar to pure Ir and to the sample containing ca. 89.5 at.% Pd.

The electrodeposits tested by the SEM technique were then characterized by means of cyclic voltammetry (Figure 3a,b). From the cyclic voltammetry curves registered in the wide potential range of −0.05 to 1.5 V, the signals originating from both hydrogen sorption and surface oxidation/reduction were noticed. The signals occurring in the potential range of −0.05 to 0.4 V were associated with hydrogen absorption (reduction; signal A) and desorption (oxidation; signal B). Additionally, the signals of hydrogen adsorption (signal A’) and desorption of adsorbed hydrogen (signal B’) were present, but noticeable only for the electrode containing 89.5 at.% Pd. It is obvious that even a small amount of Ir (ca. 0.5 at.%) influenced the change in the position of the hydrogen desorption signal. The alloying with Ir caused a shift of the hydrogen oxidation signal (signal B) into lower potential values (compared with Pd), indicating the facilitation of the process of hydrogen removal from the alloy. The shape of the hydrogen oxidation signal (signal B) for samples containing 0.5–1 at.% Ir was very similar to that obtained for Pd. This means that in these alloys, similar amounts of hydrogen were absorbed, as well as the creation of the β-phase followed in the similar potential values. The CV of the alloy containing ca. 89.5 at.% of Pd differed from the others. The position of the signal of hydrogen oxidation (signal B) was placed at a lower potential value than that for the other tested alloys. Moreover, the shape of the hydrogen oxidation and reduction signals indicated that in this alloy, only hydrogen in the α-phase was present. The CV registered for pure Ir showed that this metal was not able to absorb hydrogen, since the currents in the region of hydrogen absorption and oxidation were negligible. Figure 3b presents a zoomed image of part of Figure 3a (0.2–1.5 V) in the region of surface oxide formation (signal C) and reduction (signal D). It was noted that alloying Pd with Ir also influenced the surface state of the electrodeposited alloys. The increasing content of Ir in deposits resulted in a shift of the surface oxide reduction signal (signal D) into lower potential values compared with Pd. This indicates that there was an increase of the Ir content on the surface of the alloys.

As previously stated in the literature [13], for the electrodes deposited from the galvanic baths, quite often, the surface composition differs from the bulk one. Thus, the surface composition of the selected electrodes was measured by means of AES and compared with the bulk composition obtained by EDS. The dependence of the Pd surface content and Pd bulk content is presented in Figure 4. It can be concluded that in the case of Pd-Ir alloys, surface enrichment in Pd was observed. This means that the Ir content at the electrode surface was less than that in the bulk. This finding is in line with the results previously presented by Aas et al. [29] and Schwartz et al. [30] for Pd-Ir alloys. The effect of surface enrichment with Pd was observed for electrodeposited Pd-Au alloys, whereas the opposite effect—surface enrichment with the alloying metal—was observed in the case of Pd-Rh and Pd-Ru alloys [13]. The final state of the alloy, i.e., the bulk and the surface composition, depends on the state of the surface energy. The surface enrichment in Pd for Pd-Ir alloys is ascribed to the higher surface energy of Ir (3.05 Jm^−2^) than that of Pd (2.05 Jm^−2^) [19].

As stated above, the composition of the electrodeposited alloys can be different in the bulk and at the surface. This also depends on the technique used for its determination. In Table 2, a comparison between the composition of the Pd-Ir alloys determined with the use of four techniques, such as: AES, XPS, EDS, and AAS, is presented. The first two techniques can be used to examine the surface composition, and the other two techniques (thickness of the studied electrodes up to ca. 1 µm) to define the bulk composition. If the results of AES and XPS are compared, it should be underlined that AES yields information from a few surface layers, whereas in the case of XPS, estimation of the composition concerns ca. 10 nm (depending on the metals). The analysis of the results in Table 2 enabled us to note the good agreement between the data obtained from EDS and AAS. Since AAS is a destructive method, EDS can effectively be used to determine the bulk composition of Pd-Ir alloys. The data obtained for AES and XPS cannot be strictly compared because of the reason described above. In the case of every studied sample, the composition determined by AES indicated a higher Pd content than XPS.

Figure 5 shows the EDS maps of the Pd-Ir alloy electrode containing ca. 99 at.% Pd, presenting the distribution of Pd and Ir in the area marked with a green rectangle. The EDS maps for Pd and Ir show the bulk homogeneity of the electrode and confirmed that both elements were uniformly distributed over a large area. Moreover, from the analysis of Figure 3b, it can be stated that Pd-Ir alloys were homogenous at the surface, since in the case of every Pd-Ir alloy, one signal of surface oxide reduction could be distinguished.

### 3.3. Electrochemical Dissolution of Pd-Ir Alloys

The selected Pd-Ir alloy electrodes with different compositions were subjected to two experiments of electrochemical dissolution: one in the potential range of −0.05 to 1.5 V and the other in the potential range of 0.4 to 1.5 V. In the first experiment, the electrodes were cycled both in the region of hydrogen sorption as well as in the region of surface oxide formation and reduction. In the second experiment, the electrodes were cycled only in the region of surface oxide formation and reduction.

From the analysis in Table 3, it was concluded that in the case of the first experiment, significant changes in the electrode composition occurred. It can therefore be concluded that cycling the electrodes in the ‘hydrogen sorption’ region involved more intense dissolution of Ir from Pd-Ir electrodes, resulting in electrode enrichment in Pd. Whereas, in the case of cycling in the ‘surface oxide’ region, the dissolution of the electrodes occurred without significant changes in the composition.

Figure 6a presents the cyclic voltammetry behavior of the selected Pd-Ir alloy during the experiment of electrochemical dissolution in the potential range involving the hydrogen sorption region. The analysis of the charge connected with hydrogen desorption and surface oxide reduction (Figure 6b) allowed us to discuss the influence of the dissolution on the changes of the real surface area and the hydrogen sorption capacity. From Figure 6b, it is obvious that in the first ca. 100 cycles, the real surface area of the electrode increased, whereas from ca. 100 to 250 cycles, a decrease of the real surface area was observed. However, at the end of the experiment, the real surface area was higher than in the beginning. The comparison of these results with the ones obtained without cycling in the hydrogen region also showed the tendency to increase the real surface area; however, the drop of this value started at ca. 20 cycles. These data cannot be quantitatively compared since the charges were normalized to the value of maximum charge. The results presented in Figure 6b also confirmed enrichment of the electrodes in Pd, since cycling in the wide range of potential caused the increase of the hydrogen oxidation charge. This indicates that with every single cycle, the electrode is composed of the alloy containing more Pd. This is in line with the EDS data in Table 3. Figure 7 shows the morphology of the electrode containing ca. 92.7 at.% Pd before the electrochemical dissolution (Figure 7a), after ca. 250 cycles in the region involving both hydrogen sorption and surface formation/reduction (Figure 7b), and in the region only involving the latter process (Figure 7c). The morphologies of the electrodes subjected to the cycling in different potential regions significantly varied. The process of hydrogen sorption caused the creation of the porous structures protruding from the surface. Moreover, there were many visible cracks caused by hydrogen insertion/removal in/from electrodes. The surface of the electrode after cycling in the region where no hydrogen sorption occurred more closely resembled the one before electrochemical treatment. In this case, some cracks at the surface were also visible, however, they were connected with the stresses in deposits during continuous cycling. The comparison of the SEM image of electrodes after ca. 250 cycles and the fresh one revealed that electrode dissolution involving hydrogen sorption caused a higher surface expansion.

### 3.4. The Influence of the Hydrogen Sorption on the Surface State of the Pd-Ir Alloys

The impact of the process of single- and multiple-hydrogen sorption on the surface state of Pd-Ir electrodes was studied using electrodes with different compositions. In Figure 8, the influence of the single/multiple-hydrogen sorption and the electrode composition on the relative surface roughness factor (R/R_o_), the potential of surface oxide formation (E_ox_), and reduction (E_red_) is presented. The relative surface roughness factor (R/R_0_) was calculated through division of the surface roughness factor in some stages of the electrochemical treatment via the surface roughness factor of the freshly prepared sample. To obtain the surface roughness factor, the real surface area was divided by the geometric area of the electrode. The real surface area was appointed with the use of the charge of the surface oxides’ reduction (for details, see Equation (1) in [15]). The single-hydrogen sorption caused a negligible impact on the value of the real surface area, but only for the electrodes containing ca. 95% Pd and above. For the electrodes containing more than 5% at. Ir, there was a significant increase of the real surface area. Moreover, after multiple-hydrogen sorption, a decrease of the real surface area was observed, but only for the electrodes in the composition range of 95 to 100 at.% Pd. The interaction of the electrode with hydrogen also influenced the onset of the signals of oxide formation (surface oxidation) and oxide reduction. Significant changes in the potential values after single- and multiple-hydrogen sorption were visible in the case of surface oxide formation. After single-hydrogen sorption, the onset of the signal of surface oxidation shifted into lower potential values; then, after multiple-hydrogen sorption, the tendency of the changes was the opposite. This indicates that single exposure to hydrogen causes the facilitation of surface oxidation, whereas multiple-hydrogen sorption inhibits this process. In the case of the values of surface oxide reduction, only slight changes were visible for electrodes containing up to ca. 3 at.% Ir. After multiple-hydrogen sorption, the potential of oxide reduction shifted to higher values, indicating facilitation of the process of oxide reduction. It is noteworthy that Pd-Ir alloys in the described processes exhibited the opposite tendency to pure Pd and other types of Pd binary alloys with noble metals. In the case of Pd-Ru, Pd-Au, and Pd-Pt-Au, the facilitation of surface oxide formation and the inhibition of surface oxide reduction were observed [13]. The obtained results showed that single and multiple exposure to hydrogen changed the surface state of the electrodes.

### 3.5. The Thermodynamics and Kinetics of the Hydrogen Absorption in Pd-Ir Alloys and the Miscibility Gap in the Pd-Ir-H System

As previously stated [16], Pd-Ir alloys exhibit interesting properties in relation to their interaction with hydrogen. For the alloys containing relatively high amounts of Pd, β-phase of hydrogen was created. Alloying Pd with Ir in the case of every studied composition resulted in a capacity drop; however, for the alloys containing less than ca. 3 at.% of Ir, the capacity was comparable to pure Pd (H/Pd ca. 0.72) [16]. In Figure 9a, the dependence of the transition potential of hydrogen absorption and desorption on the Pd bulk content in acidic medium is visible. It can be noted that both the values of α → β and β → α decreased with the increase of the Ir content. This phenomenon is well-known in the electrochemical characteristics of Pd-binary alloys and constitutes the electrochemical evidence for the formation of contracted alloys. In our previous research [16], this was also confirmed by structural studies. Figure 9a also indicates that alloying Pd with increasing amounts of Ir resulted in a decrease of the hysteresis extent. It can be estimated from Figure 9a that the hysteresis vanished for the alloys containing ca. 6 at.% Ir. This showed that for the alloy containing ca. 6 at.% Ir, only a solid solution of hydrogen (α-phase) was formed in the alloy. However, this method only allows for an approximate determination of the composition. For an accurate determination of the critical composition, a calculation of the maximum concentration of hydrogen in the α-phase and the minimum concentration of hydrogen in the β-phase was performed. The values were calculated based on the integration of the specific parts of the chronoamperometric curves registered at the 0.48 V hydrogen desorption potential and at the −0.016 V hydrogen absorption potential [27]. The above calculations revealed that formation of the β-phase did not occur for the alloy containing ca. 93.7 at.% Pd (the miscibility gap vanished for the Pd-Ir alloy containing ca. 93.7 at.% Pd). This value is in line with that obtained on the basis of the dependence of the transition potentials on the Pd bulk content.

Figure 10a presents chronopotentiometry curves for the process of hydrogen absorption and desorption in Pd and selected Pd-Ir alloys in acidic and alkaline media. It can be noted that alloying Pd with Ir resulted in a slight drop of capacity, and the potential of α → β/β → α was shifted to lower values, indicating that Pd-Ir alloys can be classified as contracted alloys. The electrolyte used as a source of hydrogen, i.e., protons from acidic medium and the water molecule in alkaline medium, did not have much influence on the potential of α → β phase transition. The slight deviations were rather the effect of some impurities [27] present in the strong alkaline medium, influencing the values of the transition potential. From the position of the plateau in the absorption and desorption curves, it is also possible to discuss the hysteresis effect, which was smaller in the case of Pd-Ir alloys in both the acidic and alkaline media. This was also observed in the case of other Pd binary alloys [27]. The presented results indicate that an acidic medium as well as a strong alkaline medium can be effectively used as electrolytes in the process of hydrogen sorption in Pd-Ir alloys, since the type of electrolyte did not quantitatively influence the process of hydrogen sorption (capacity, transition potential). The presence of Ir influenced the kinetics of the hydrogen sorption process. In Figure 10b, it is shown that alloying with Ir caused an increase of the intensity of the maximum current of hydrogen absorption. In the case of the alkaline medium, a reduction of the absorption time was also observed for Pd-Ir alloys, compared to Pd. Moreover, in the alkaline electrolyte in the case of Pd and Pd-Ir alloys, a decline of the maximum current intensity, and prolongation of the absorption time, were observed compared to the acidic medium. The described results showed that alloying Pd with Ir facilitated the process of hydrogen sorption, whereas it was impeded in the strong alkaline medium. In summary, hydrogen sorption in alkaline electrolytes did not have much influence on the hydrogen sorption thermodynamics, but it had a substantial influence on the kinetics of the studied process.

The experiments of hydrogen sorption performed at different temperatures can be used to calculate the thermodynamic functions of hydride formation/decomposition. As stated previously, since there can be some variation of the transition potential in alkaline media, for the calculation of thermodynamic functions, the results from the acidic medium were utilized. The potential of the α → β phase transition can be strictly recalculated into Gibbs energy. The dependence of the Gibbs energy on the temperature enabled to calculate values of enthalpy and entropy of hydride formation/decomposition. With the increasing Ir content, the values of Gibbs energy of hydrogen absorption increased, indicating a weaker stability of the hydride phase. In Figure 11, values of ΔH_α→β_ and TΔS_α→β_ for the process of hydride formation are presented. The value of ΔH_α→β_ in Pd-Ir alloys slightly increased with the increasing Ir content, which implies a decreasing heat of hydride formation. Slight variation of the values of ΔS_α→β_ with the Ir content was also observed. The analysis of the changes of ΔS_α→β_ values with the alloys’ composition in the studied composition range yielded information about how hydrogen atoms occupy both Pd- and Ir-neighboring interstices positions. In Figure 11, the values of the thermodynamic functions for other kinds of binary Pd alloys obtained by electrodeposition are presented, such as: Pd-Ru, Pd-Pt, and Pd-Rh alloys [12,27]. It is worth noting that from the thermodynamic point of view, Pd-Ir alloys are the most similar to Pd-Pt alloys.

## 4. Conclusions

A variety of Pd-Ir alloys with different compositions were successfully electrodeposited from the aqueous chloride baths, with a current efficiency of ca. 90%. It was found that the decrease of the deposition potential more strongly influenced the increase of the Ir content in the alloy compared to the concentration of the Ir ions in the deposition bath. The comparison between the different methods of composition estimation (EDS, AAS, AES, XPS) revealed surface enrichment of Pd-Ir alloys in Pd. Electrodeposited Pd-Ir alloys exhibited homogenous surfaces and bulk composition. It was noted that alloying Pd with even ca. 0.5 at.% Ir strongly changed the morphology of the electrodes. The dissolution of the electrodes in the potential range involving hydrogen sorption caused a decrease of the Ir content in the electrodes and a higher surface expansion compared to electrodes cycled only in the region of surface oxide formation/reduction. It was also noted that single- and multiple-hydrogen sorption had different influences on the surface state of the Pd-Ir alloys, i.e., the surface roughness factor, the onset potential of oxide formation, and the potential of oxides’ reduction. The study of the influence of the electrolyte on the hydrogen sorption in Pd-Ir alloys revealed that the type of electrolyte did not significantly influence the thermodynamics of hydrogen sorption, whereas it had a substantial impact on its kinetics. The critical composition of the Pd-Ir alloys, where the miscibility gap of hydrogen in α- and β-phase in Pd-Ir alloys vanished, was determined at a composition of ca. 93.7 at.% Pd. Pd-Ir alloys in the form of thin layers can be applied to modify the anode material (e.g., AB_5_ alloys) used in Ni-MH batteries. In future research, the utilization of Pd-Ir alloys in the form of nanoparticles can be used in the decoration of the surface of AB_5_ hydrogen storage alloys [31]. Since in this study, the Ir-doping of Pd resulted in the facilitation of the hydrogen sorption process, it is also expected that the surface modification of AB-type hydrogen storage alloys with Pd-Ir alloys will enhance the kinetics of the hydrogen sorption process in Ni-MH batteries.

## Figures and Tables

**Figure 1 materials-16-04556-f001:**
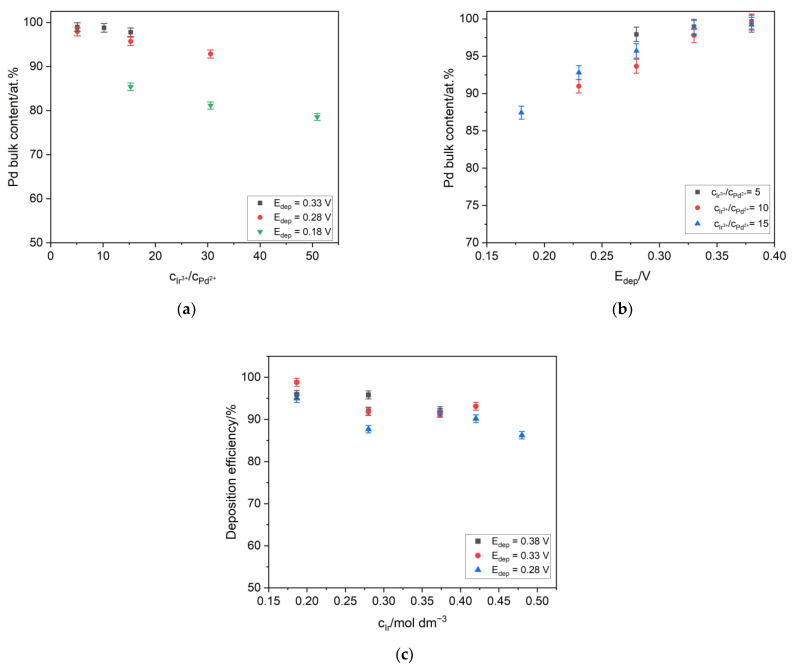
(**a**) The influence of the Ir to Pd ions’ concentration ratio (in the galvanic baths) on the Pd bulk content. (**b**) The influence of the potential deposition on the Pd bulk content. (**c**) The influence of the Ir ions’ concentration on the current deposition efficiency.

**Figure 2 materials-16-04556-f002:**
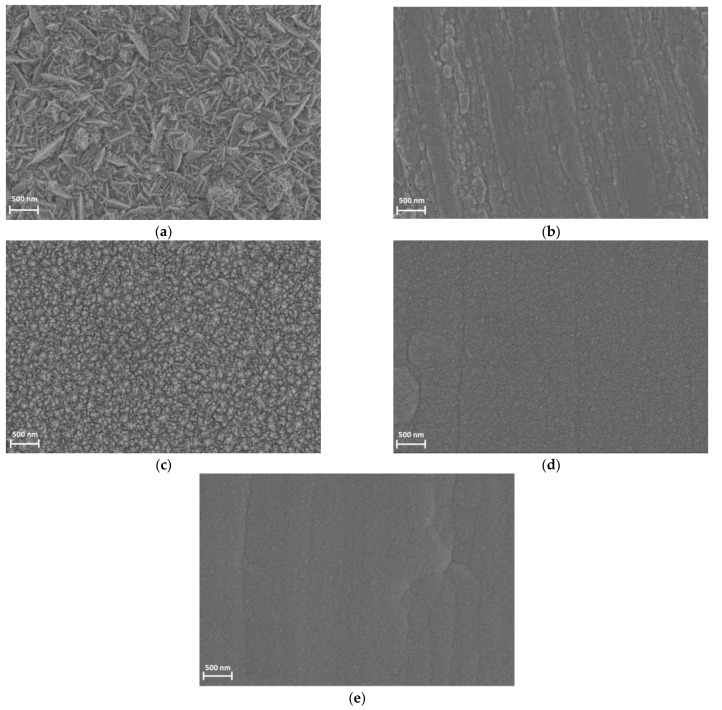
The SEM images of the electrodeposited Pd-Ir alloys: (**a**) 100 at.% Pd, (**b**) 100 at.% Ir, (**c**) 99.5 at.% Pd, (**d**) 99.0 at.% Pd, and (**e**) 89.5 at.% Pd.

**Figure 3 materials-16-04556-f003:**
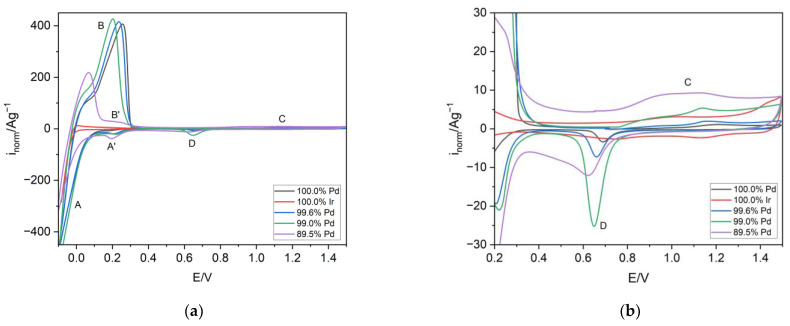
(**a**) The cyclic voltammograms for Pd, Ir, and Pd-Ir alloys of selected composition in 0.5 M H_2_SO_4_ in the potential range of −0.05 to 1.5 V, scan rate 100 mVs^−1^. (**b**) Zoomed image of part of Figure 3a in the potential range 0.2–1.5 V.

**Figure 4 materials-16-04556-f004:**
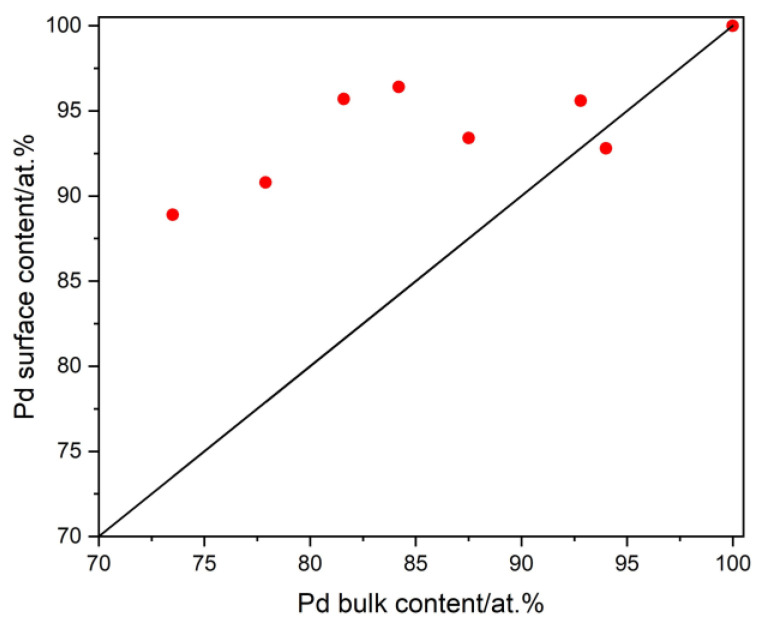
The relation between the Pd surface composition (determined by AES) and the Pd bulk composition (determined by EDS). The solid black line expresses the relation when surface composition = bulk composition.

**Figure 5 materials-16-04556-f005:**
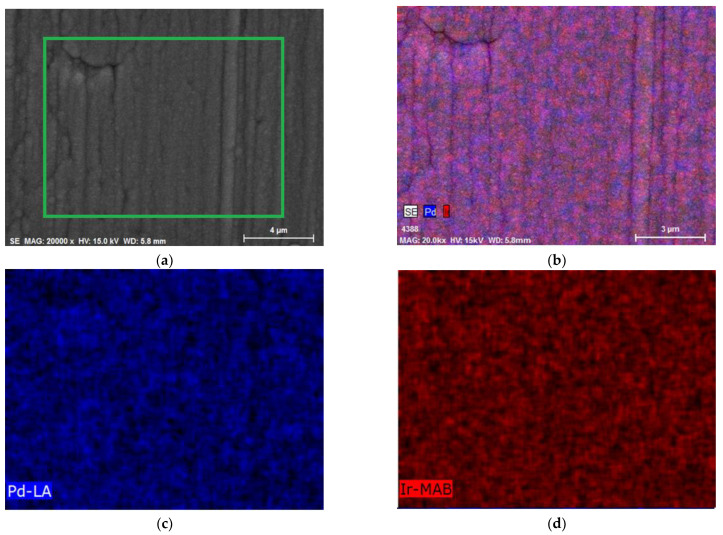
EDS maps for Pd-Ir alloys containing ca. 99 at.% Pd. (**a**) SEM image: EDS maps registered for the area marked with the green rectangle. (**b**) Collective EDS map for Pd and Ir. (**c**) EDS map for Pd and (**d**) EDS map for Ir.

**Figure 6 materials-16-04556-f006:**
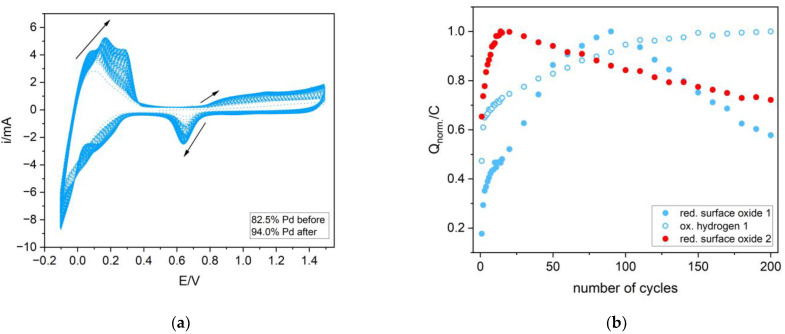
(**a**) The cyclic voltammogram of the Pd-Ir alloy containing ca. 82.5 at.% Pd before the last 200 cycles of electrochemical dissolution and 94.0 at.% after electro-dissolution in 0.5 M H_2_SO_4_ in the potential range of −0.1 to 1.5 V, scan rate: 100 mVs^−1^. (**b**) The dependence of a normalized charge on the number of cycles. Blue closed symbols (red. surface oxide 1)—charge connected to the reduction of surface oxide during electrode dissolution in the −0.1–1.5 V potential range, open blue symbols (ox. hydrogen 1)—charge connected to hydrogen oxidation during electrode dissolution in the −0.1–1.5 V potential range, and red closed symbols (red. surface oxide 2)—charge connected to the reduction of surface oxide during electrode dissolution in the 0.38–1.5 V potential range.

**Figure 7 materials-16-04556-f007:**
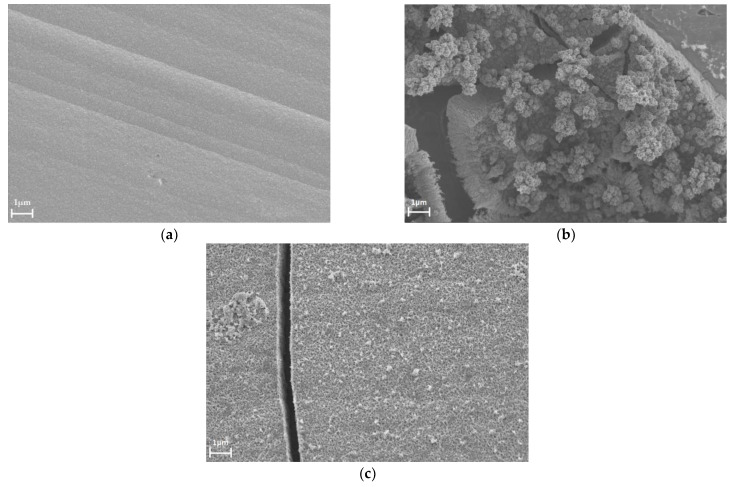
The SEM images of the electrodeposited Pd-Ir alloy: (**a**) before electrochemical dissolution—alloy composition: 92.7 at.% Pd, (**b**) after electrochemical dissolution in a wide potential range (−0.1–1.5 V)—alloy composition ca. 99.6 at.% Pd, and (**c**) after electrochemical dissolution in the ‘surface oxide’ region—alloy composition ca. 93.1 at.% Pd.

**Figure 8 materials-16-04556-f008:**
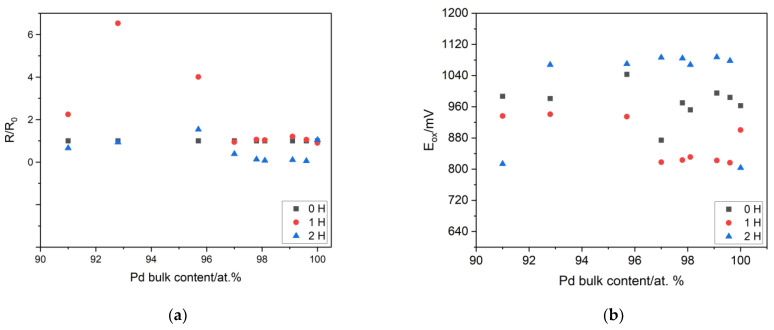
The influence of the Pd bulk content on: (**a**) the relative value of the surface roughness factor (surface roughness factor normalized to the surface roughness factor of the freshly deposited electrode), (**b**) the onset of the potential of surface oxidation, and (**c**) the potential of surface oxide reduction. 0 H: freshly deposited electrode, 1 H: electrode after single-hydrogen sorption, 2 H: the electrode after multiple-hydrogen sorption, electrolyte: 0.5 M H_2_SO_4_.

**Figure 9 materials-16-04556-f009:**
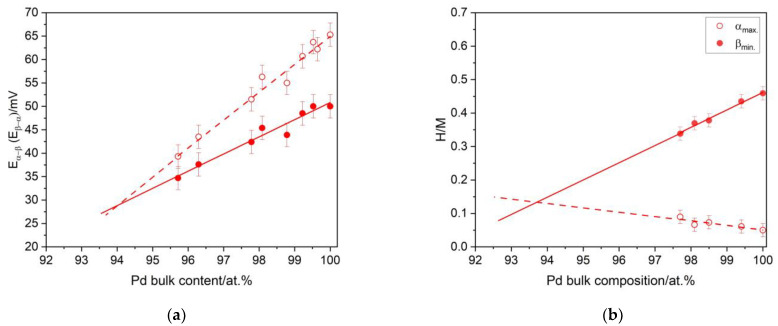
The influence of the alloy bulk composition of Pd-Ir alloys on: (**a**) the potential of β → α (open symbols) and α → β (closed symbols) phase transition, and (**b**) the maximum hydrogen concentration in the α-phase and the minimum hydrogen concentration in the β-phase.

**Figure 10 materials-16-04556-f010:**
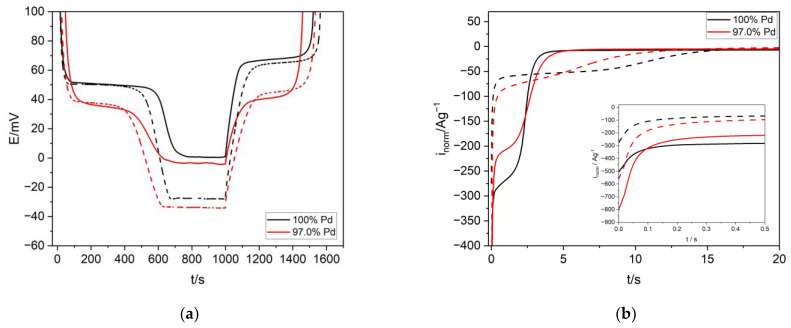
(**a**) The chronopotentiometry dependence of the hydrogen charging and discharging process in Pd and selected Pd-Ir alloys (i= 1.0 A·g^−1^) in 0.5 M H_2_SO_4_ (solid line) and 6 M KOH (dashed line). (**b**) Chronoamperometry dependence of hydrogen absorption in Pd and selected Pd-Ir alloys.

**Figure 11 materials-16-04556-f011:**
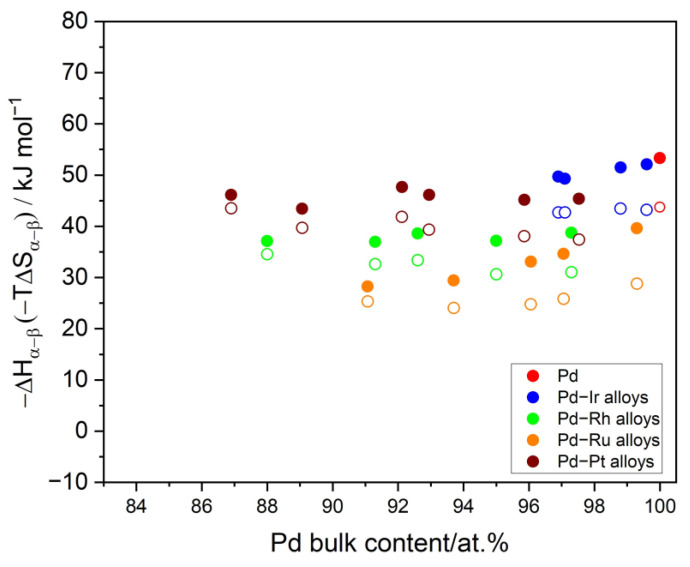
The influence of the Pd-Ir alloy bulk composition on the values of thermodynamic functions of hydrogen absorption (∆H_α→β_: closed symbols, T∆S_α→β_: open symbols, calculations for T = 298 K).

**Table 1 materials-16-04556-t001:** Properties of the palladium and iridium [24].

Metal	Atomic Radius (Å)	Lattice Constant (Å)	Melting Points (°C)
Pd	1.37	3.883	1554
Ir	1.35	3.831	2454

**Table 2 materials-16-04556-t002:** Pd-Ir alloys’ compositions determined with the use of different techniques. E_dep_: deposition potential (E vs. SHE), c_Ir_/c_Pd_: the Ir to Pd ions’ concentration ratio.

E_dep_ (V)	c_Ir_/c_Pd_	Pd Content/at.%			
		AES	XPS	EDS	AAS
0.28	15.3	92.8	92.3	94.0	96.4
0.18	15.3	93.4	93.2	87.5	85.4
0.28	30.6	95.6	94.5	92.8	92.8
0.18	30.6	96.4	95.9	84.2	81.1
0.13	30.6	90.8	88.5	77.9	70.0
0.18	50.9	95.7	91.4	81.6	78.6
0.13	50.9	88.9	83.3	73.5	72.0

**Table 3 materials-16-04556-t003:** The influence on the number of cycles and the range of potential values on the composition of the alloys (EDS). The number with ’ relates to the electrochemical dissolution (number of cycles) in the potential range of surface oxide formation/reduction.

Number of Cycles									
	0	5	5′	15	15′	50	50′	250	250′
Pd bulk content/at.%	94.0	95.4	94.9	96.5	95.2	97.1	95.4	100	95.7
	86.2	86.7	86.3	88.0	86.5	87.8	86.7	94.1	86.7
	92.7	94.2	93.3	94.9	93.0	95.0	93.1	99.6	93.1
	83.4	84.6	84.2	85.0	83.9	86.2	84.3	99.2	84.3
	76.4	77.4	83.2	77.1	80.3	78.6	79.8	94.5	76.8
	80.7	79.6	80.4	80.6	78.2	82.5	78.4	94.0	81.4
	74.3	73.4	75.4	73.2	75.9	73.4	74.0	80.8	74.0

## Data Availability

The data presented in this study are available upon request from the corresponding author.

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
