# Peer review of "The Effect of the Iridium Alloying and Hydrogen Sorption on the Physicochemical and Electrochemical Properties of Palladium"

_materials, 2023, doi:10.3390/ma16134556_

Round 1

Reviewer 1 Report

The manuscript demonstrates the hydrogen adsorption ability of different Ir-Pd alloy composite materials. The results could be helpful for implementations in the electrochemical domain. The data presentation looks good; however, the authors should carry out a few of the comments given below to revise the article.

Comments
1. The author should represent the XRD data for the Ir-Pd alloys, author could get information about phase identification of Ir-Pd alloys, electronic shift, which could be good evidence for the interaction between iridium and palladium. which would enhance the hydrogen adsorption to support the author's claims.
2. In general, palladium has good hydrogen adsorption, in the context of the formation of an iridium alloy with palladium, the author should represent the XPS data for the oxidation state of iridium as well as palladium, and it needs to be corelate with hydrogen adsorption.
3. There are so many articles reported about hydrogen adsorption using valuable metal alloys that the author should explain why used Pd-Ir alloy.
4. What is the reason behind the higher hydrogen sorption on Ir-Pd alloys compared to neat?
5. It is suggested that the authors explain the hydrogen adsorption mechanism.

Reviewer 2 Report

1. There are large volume of studies published on Pd and Ir nanoparticles for various applications. It is suggested to cite them and explain briefly. Please see: (https://doi.org/10.1016/j.cej.2022.140289, ACS Applied Materials & Interfaces 15 (12), 16177-16188, J. Phys. Chem. C 2023, 127, 20, 9594–9602).

2. Table 1 is suggested to move in supplementary information or write in text form instead of tabular form.

3. The ligands of all Figures should be on top left corner.

4.  The different peaks in CV profile (Figure 3) should me marked and explained properly.  Please see: ACS Applied Energy Materials 4 (12), 14043-14058

Minor spelling mistakes should be corrected.

Reviewer 3 Report

The authors presented the results of the physiochemical and electrochemical studies, as well as surface/bulk properties of the Pd-Ir alloys in the presence of the hydrogen sorption.

I found the authors did a lot of jobs, however, the presentation and organization of the results are very confusing and my impression is that difficult to get the most interesting points of the experiments. The authors must revise the manuscript in the context that they choose a more appropriate alloy composition (or a maximum of three alloys) and emphasize the most important point and conclusion of the investigation. Also, some strong conclusion is missing, giving a lot of space for speculation. In addition, some conclusions are not supported by relevant experimental results.

In detail:

In Figure 2. The SEM images must be given in order of the Pd percentage!

line 192: "It means that the content of the Ir on the surface of the electrode……."

The authors give the implicit conclusion based on the content of the Pd. Why they did not determine Ir content in the bulk and the surface?

In fact, the authors presented results (Figure 5) of the EDS for 99 % of the Pd, although Figure 4 reveals that for this electrode there is no difference in the Ir concentration at the surface and in the bulk.

Accordingly, the conclusion is lack arguments.

In Table 3 there is column "0". How is data from this column relate to the data from Table 2? I found some connections, but mostly the presented data is different.  In fact what data are given in column "0"?

In figure 6. is presented alloy containing 82.5 % Pd?! This percentage is related to the what row in Table 3 or Table 2?

In Figure 8. "relative real surface area" is given. How this area is determined?

Finally, the thermodynamics and kinetics data, obtained in such an unstable system are not so reliable. Under an unstable system, I meant that the electrode surface as well as composition is changing during electrolysis and time.

The conclusion being presented in lines 357-359 is too audacious when we bear in mind experimental data.

And finally, a conclusion concerning the applicability of these alloys for the purpose mentioned in the Introduction as well as some suggestions for further investigations, improvement, etc should be given.

Round 2

Reviewer 3 Report

The authors were response in detail on my comment and they significantly  improve the manuscript in accordance with reviewer suggestions.

I found the article suitable for publishing in present form.

Best regards